



# Hygroscopic Aerosols Amplify Longwave Downward Radiation in the Arctic

Denghui Ji[1,*], Mathias Palm[1], Matthias Buschmann[1], Kerstin Ebell[2], Marion Maturilli[3], Xiaoyu Sun[1], and Justus Notholt[1]

[1]Institute of Environmental Physics, University of Bremen, Otto-Hahn-Allee 1, 28359 Bremen, Germany
[2]Institute for Geophysics and Meteorology, University of Cologne, Zülpicher Str. 49a, DE-50674 Cologne, Germany
[3]Alfred Wegener Institute, Helmholtz Centre for Polar and Marine Research, Telegrafenberg A43, 14473 Potsdam, Germany

**Correspondence:** Denghui Ji (denghui_ji@iup.physik.uni-bremen.de)

**Abstract.** This study investigates the impact of hygroscopic aerosols, such as sea salt and sulfate, on longwave downward radiation in the Arctic. These aerosols absorb atmospheric water vapor, leading to wet growth, increased size, and enhanced longwave downward radiation emission, defined as the Aerosol Infrared Radiation Effect. Observations of aerosols, especially their composition, are challenging during the Arctic winter. We use an emission Fourier Transform Spectrometer to measure

aerosol composition. Observations show that the Aerosol Infrared Radiation Effect of dry aerosols is limited to about $1.45 \pm 2.00$ $\mathrm{Wm}^{-2}$. Wet growth significantly increases this effect. During winter, at relative humidity levels between 60% and 80%, wet aerosols exhibit effects approximately 10 times greater than dry aerosols. When relative humidity exceeds 80%, the effect can be up to 50 times higher (30 - 100 $\mathrm{Wm}^{-2}$). Sea salt aerosols in Ny-Ålesund demonstrate high effect values, while non-hygroscopic aerosols like black carbon and dust show consistently low values. Reanalysis data indicates increased water vapor

and sea salt aerosol optical depth in Ny-Ålesund after 2000, correlating with significant positive temperature anomalies in this area. Besides, wet aerosols can remain activated even in dry environments, continuously contributing high effects, thereby expanding the area affected by aerosol-induced warming. This warming effect may exacerbate Arctic warming, acting as a positive feedback mechanism.

## 15 1 Introduction

Arctic amplification (AA), characterized by the accelerated warming of the Arctic region compared to global averages, is a phenomenon of importance in climate change (Serreze and Barry, 2011; Wendisch et al., 2017; Peace et al., 2020). This amplified warming is particularly pronounced during the polar night, highlighting the need for a comprehensive understanding of its causes and consequences (Chung et al., 2021). To elucidate the underlying mechanisms driving AA, extensive research

has focused on key processes such as temperature feedback, surface albedo feedback, and cloud and water vapor feedback (Bony et al., 2006; Soden and Held, 2006; Graversen et al., 2014; Taylor et al., 2013; Philipp et al., 2020). Among these,



aerosols are an important factor in Arctic climate dynamics, influencing various feedback mechanisms. For instance, dust and Black Carbon (BC) deposition on snow or ice surfaces reduce albedo, accelerating ice melt (Ming et al., 2009; Bond et al., 2013). Moreover, sea salt aerosols modify cloud properties, enhancing longwave downward radiation (LWD) and contributing
to surface warming (Gong et al., 2023).

In the context of the Arctic energy budget, LWD constitutes a critical component, primarily governed by greenhouse gases (GHGs) in the global mean (Trenberth et al., 2009; Wild et al., 2015; Tian et al., 2023). However, during the polar night when solar shortwave radiation is absent, the cooling effect of clouds and aerosols, arising from the scattering of solar radiation, becomes negligible (Cox et al., 2015). Therefore, LWD from clouds and aerosols assumes greater significance, particularly in
maintaining the Arctic energy balance (Cox et al., 2015; Serreze and Barry, 2011; Lenaerts et al., 2017; Ebell et al., 2020). Compared with LWD from clouds ( 50 - 100 $\mathrm{Wm}^{-2}$ (Cox et al., 2015; Serreze and Barry, 2011; Lenaerts et al., 2017; Ebell et al., 2020)), previous studies show that the LWD caused by aerosols in dry condition is usually lower than 10 $\mathrm{Wm}^{-2}$ (Spänkuch et al., 2000; Markowicz et al., 2003; Vogelmann et al., 2003; Lohmann et al., 2010). Dry aerosol particles contribute very limited LWD to the Arctic climate. However, LWD in the transition state (wet aerosols) between dry aerosols and cloud
droplets is rarely mentioned.

Aerosols in the atmosphere, including sea salt and sulfates, possess hygroscopic properties, allowing them to absorb water vapor and undergo wet growth (Winkler, 1973). This process, known as aerosol wet growth, is accompanied by an increase in LWD (Mauritsen et al., 2011). The magnitude of this increase is influenced by factors such as aerosol composition, and ambient relative humidity (RH) (Peng et al., 2022). Notably, the deliquescence point, at which hygroscopic aerosols abruptly
increase in size, is a critical threshold determined by ambient RH (Tang and Munkelwitz, 1993; Winkler, 1973). For example, the sea salt and sulfate aerosols in this study have deliquescence points of about 75% and 85%, respectively (Peng et al., 2022). This means that when the ambient humidity increases to 75%, the dry sea salt aerosol particles can absorb water vapor in the atmosphere and become larger. If the ambient humidity continues to increase, the sea salt wet particles will continue to absorb water and become sea salt solution droplets (still belonging to aerosols). This is the wet growth process of sea salt aerosols.
Recent studies have shown an increase in Arctic water vapor content, attributed to enhanced poleward transport facilitated by atmospheric river pathways (Sato et al., 2022; Thandlam et al., 2022; Bresson et al., 2022; Lauer et al., 2023). Moreover, sea salt aerosols have been identified as dominant contributors to Arctic aerosol composition during the winter season (Huang and Jaeglé, 2017; Kirpes et al., 2018). Therefore, the rise in coarse-mode aerosols, primarily originating from sea spray, and the increase of RH in the Arctic underscore the need to investigate the potential impact of aerosols on LWD and Arctic warming
during their wet growth process (Heslin-Rees et al., 2020; Pernov et al., 2022).

Given the complex interplay between aerosols (especially aerosol composition), RH, and LWD, understanding the radiative effects of aerosol wet growth is crucial for understanding the role of aerosols in AA, particularly during the polar night. Considering the various factors contributing to atmospheric LWD, such as greenhouse gases, clouds, and aerosols, this study aims to explore the extra LWD introduced during aerosol wet growth. Thus, this study focuses on hygroscopic aerosols, particularly
sea salt and sulfate aerosols. To achieve this, both model simulations and observational data (Site location: Ny-Ålesund; Time period: Dec-Jan-Fer, 2017- 2022) will be utilized, defining the resulting additional LWD from aerosols as the Aerosol Infrared





Radiation Effect (ARE). This paper is structured as follows: Section 3 provides an overview of the datasets utilized, including Fourier-transform infrared spectroscopy (FTS) and Baseline Surface Radiation Network (BSRN) measurements. Section 4 outlines the methodologies employed to derive ARE from LWD measurements (eliminating contributions from clouds and GHGs) in detail. The results are presented in Section 5. Finally, the implications of these findings are discussed in the conclusion section 7.

## 2 Site Description

Ny-Ålesund (78.925 °N, 11.925 °E), Svalbard, is located in the North Atlantic atmospheric transport gateway to the Arctic. It serves as a central hub for international Arctic research, attracting scientists from around the world to study various environmental and climate-related phenomena, in particular to monitor Arctic amplification. The region stretching from Svalbard to the Barents and Kara Seas is currently experiencing particularly intense winter warming, with temperatures rising by more than +3 K per decade (Dahlke and Maturilli, 2017). This region is also an important pathway for air mass transport between the Arctic and mid-latitudes (Graßl et al., 2022). Today, Ny-Ålesund is primarily a research town, hosting several year-round research stations operated by different nations. Key activities focus on long-term atmospheric monitoring, studying the effects of climate change in the Arctic, and tracking the transport of pollutants and aerosols from lower latitudes to the Arctic. Due to its location and concentration of research infrastructure, Ny-Ålesund is a well-known site for Arctic research, providing invaluable data and insights into one of the most rapidly changing regions on Earth.

## 3 Data

All observations and model simulations in this study are conducted in Ny-Ålesund. The surface radiation measurements, radiosonde launches, and all measurements by cloud radar, microwave radiometer, ceilometer, and FTS, are operated at the Atmosphere Observatory of the AWIPEV research base that is run jointly by the German Alfred Wegener Institute, and the French Polar Institute. Data from both observations are filtered by using Cloudnet data to ensure cloud-free conditions, focusing solely on aerosols.

The FTS plays a crucial role in elucidating the relationship between aerosol composition and ARE, offering detailed insights into aerosol composition while quantifying ARE. However, the ARE from the FTS is restricted to the atmospheric window region (AW, 690 - 1390 $cm^{-1}$; 7 - 14 μm). On the other hand, the BSRN provides LWD data across the entire mid-infrared region (4.5 - 42 μm) but cannot characterize aerosol composition. Each dataset presents different advantages and limitations. Furthermore, it is essential to account for the influences of other radiative sources, such as clouds and greenhouse gases, to accurately assess ARE.





## 3.1 Clouds and Aerosols Signals from Cloudnet

In order to identify cloud cases, the Cloudnet classification product is used (Illingworth et al., 2007). Cloudnet is operationally applied to the AWIPEV measurement (Nomokonova et al., 2019; Ebell et al., 2023). Within the Cloudnet processing, information from a cloud radar, ceilometer, microwave radiometer and output from a numerical weather prediction model is combined and the backscattered signals by the radar and ceilometer are classified in terms of the occurrence of "Aerosol & insects", "Insects", "Aerosols", "Melting & droplets", "Ice & droplets", "Ice", "Drizzle & droplets", "Drizzle or rain" and "Droplets". The classification profiles have a vertical resolution of 20 m and extend from 120 m to about 11 km height above the surface. The Cloudnet data used in this study are measured from 2017 to 2022, with temporal resolution of 30s. The application of this data to the FTS and BSRN is slightly different, and the specific methods are given in the respective sections (Sec.4.1 for FTS and Sec.4.4 for BSRN).

## 3.2 LWD in Atmospheric Window Measured from FTS

A Fourier Transform spectrometer, called NYAEM-FTS, for measuring down-welling emission in the thermal infrared was installed in Ny-Ålesund in the summer of 2019. The NYAEM-FTS consists of a Bruker Vertex 80 Fourier Transform Spectrometer, an SR800 blackbody, an automatically operated gold mirror to select the radiation source, and an automatically operated hutch which shields the instrument from the environment. It is situated in a temperature-stabilized laboratory, at about 21 - 25 °C. The beamsplitter is a KBr beam splitter and the detector is an extended Mercury Cadmium Telluride (MCT) detector.

Therefore, the infrared spectra are measured by FTS. Since the infrared emission of aerosols is primarily concentrated in the atmospheric window (Ji et al., 2023), integrating the spectrum within this region provides the longwave radiation (LWD) data from the FTS. The FTS spectra used in this study are measured from 2019 to 2022. More details on the emission FTS can be found in Ji et al. (2023). The methods used to obtain the ARE from measured spectra are presented in Sec.4.1.

## 3.3 Aerosol Composition Data from FTS

Since aerosol composition should also be considered during the aerosol wet growth process, it is worthwhile to study the ARE with different aerosol compositions. Ji et al. (2023) have shown previously that the aerosol composition (sulfate, sea salt, dust, and BC) can be retrieved from emission FTS using the retrieval algorithm, called the second version of Total Cloud Water RETrieval (TCWRET-V2; TCWRET-V1 is developed by Richter et al. (2022) for cloud retrieval.). In this retrieval algorithm, the meteorological data are used and taken from ERA5 hourly data on pressure levels (Hersbach et al., 2023). In this study, the look-up tables of aerosol optical properties required for the retrieval algorithm have been updated, including the wet growth process of aerosols. Following the method described in Ji et al. (2023), sulfate (dry or wet state), sea salt (dry or wet state), dust, and BC are retrieved under different RH conditions. The retrieved aerosol composition data are from 2019 to 2022. Further details on how to retrieve aerosol composition considering aerosol wet growth are given in Sec.4.3.



### 3.4 LWD in Mid-infrared Range from BSRN

The radiation measurements (the LWD) are from Maturilli (2020) at station Ny-Ålesund. The Baseline Surface Radiation Network (BSRN) is a global network of high-quality ground-based stations established to observe amongst others the upward and downward long-wave radiation. All data are quality-controlled. LWD (4.5 - 42 μm) measurements from BSRN are expected
to have an uncertainty within $\pm 5\ \mathrm{Wm^{-2}}$ (Maturilli et al., 2015). The LWD data used in this study are measured in every winter (December-January-February, DJF) from 2017 to 2022, with a temporal resolution of 1 min.

### 3.5 Water vapor Profiles from Radiosonde

As we mentioned before, LWD measured by BSRN includes the emitted radiation of GHGs, clouds, and aerosols. Cloud cases can be identified by Cloudnet, while the contribution from GHGs should also be considered. The vertical profiles of
temperature, pressure, and RH (water vapor) are used from the radiosonde measurements (Maturilli and Dünschede, 2023). The Alfred Wegener Institute (AWI) has been performing radiosonde measurements at Ny-Ålesund since 1991, with regular daily 12 UTC launches since 1992. In order to extend this existing homogenized data record, the 2017 to 2022 Ny-Ålesund radiosonde data processed by the Global Climate Observing System (GCOS) Reference Upper-Air Network (GRUAN) have been interpolated on the according height resolution. The combined uncertainty given by the manufacturer is 4% for RH. The
duration for the radiosonde ascent profile from the surface to 30 km is about 90 minutes. The radiosonde data in this study are measured in every winter (DJF) from 2017 to 2022, with temporal resolution of 1d.

### 3.6 Reanalysis Datasets

This study uses two reanalysis datasets, one from European Centre for Medium-Range Weather Forecasts (ECMWF) Reanalysis v5 (ERA5) and the other from Modern-Era Retrospective analysis for Research and Applications version 2 (MERRA-2).
The RH and temperature data are from ERA5 monthly averaged data on pressure levels (900 hPa) from 1980 to 2022 (Hersbach et al., 2023).

The sea salt aerosol AOD data is derived from monthly MERRA-2 datasets (single level) (Gelaro et al., 2017). MERRA-2 is the latest version of global atmospheric reanalysis for the satellite era produced by the NASA Global Modeling and Assimilation Office (GMAO) using the Goddard Earth Observing System Model (GEOS) version 5.12.4. The dataset covers
the period of 1980-present. Aerosols in MERRA-2 are simulated with a radiatively coupled version of the Goddard Chemistry, Aerosol, Radiation, and Transport model (GOCART). GOCART treats the sources, sinks, and chemistry of 15 externally mixed aerosol mass mixing ratio tracers, including sea salt (Randles et al., 2017).



## 4 Methods

### 4.1 ARE$_{\text{AW}}$ from FTS

The downwelling radiance emitted by the atmosphere, including aerosols or clouds, can be measured using FTS. For Emission FTS, the waveband that is sensitive to aerosols is the atmospheric window region. To distinguish the LWD and ARE measured by FTS (7 - 14 µm) from the later mentioned LWD and ARE from BSRN (4.5 - 42 µm), here we use the subscript 'AW' to denote the quantity measured by the FTS. Note that AW is only part of the mid-infrared band of BSRN, so the radiation measured by FTS is not comparable to the radiation measured by BSRN. Additionally, in the atmospheric window, the contribution from

greenhouse gases (GHGs) is much smaller than that from clouds or aerosols, making the cloud signal the only factor that needs to be considered. The FTS conducts vertical observations, so ensuring vertical cloud-free conditions is sufficient. Specifically, when Cloudnet (Ebell et al., 2023) indicates an aerosol-only signal in the total atmospheric column, the spectra from the FTS observations for that period will be used, while spectra from other periods will be discarded. As Cloudnet provides aerosol height information, the RH at the aerosol layer is obtained from ERA5 hourly data on pressure levels (Hersbach et al., 2023),

with the error of RH about 2% (Gamage et al., 2020).

In order to calculate ARE$_{\text{AW}}$, the radiance measured by emission FTS has to be first considered to the broadband LWD$_{\text{AW}}$. The ARE in the atmosphere window is given by:

$$\text{ARE}_{\text{AW}} = \text{LWD}_{\text{AW}} - \text{LWD}_{\text{AW\_clear}} \tag{1}$$

where LWD$_{\text{AW}}$ is the calculated LWD in AW range with the measurements of emission spectra by FTS; LWD$_{\text{AW\_clear}}$ is the

emission flux from a clear atmosphere, which can be calculated using the LBLDIS model or observed by FTS under the ideal conditions of an environment without aerosols and clouds. Here, LBLDIS model simulations under a clear sky are used. The temperature, water vapor, and pressure profiles from ERA5 are used in LBLDIS as input files, other GHGs are fixed in the model. The equation for LWD$_{\text{AW}}$, from the spectral radiance $I$ (in $\text{Wm}^{-2}\text{cm}^{-1}\text{sr}^{-1}$) is given as follows:

$$\text{LWD}_{\text{AW}} = \iiint I(\upsilon, \mu, \phi) \mu \, d\mu \, d\phi \, d\upsilon \tag{2}$$

where $I$ is the radiance, $\mu$ is the cosine of the zenith angle, $\phi$ is the azimuthal angle, and $\upsilon$ is the wave number. Integrating the radiance over both the hemisphere and the wave number yields the LWD$_{\text{AW}}$. The wave number for AW ranges from 690 to 1390 $\text{cm}^{-1}$ (7 - 14 µm) (Cox et al., 2015). Similar to the method used in Cox et al. (2012), the relationship between radiance and LWD$_{\text{AW}}$ is calculated using an exponential function assumption of radiance dependence on $\mu$, as follows:

$$\int_{690cm^{-1}}^{1390cm^{-1}} I(\upsilon, \mu) d\upsilon = \int_{690cm^{-1}}^{1390cm^{-1}} I(\upsilon, \mu = 1) \cdot (a \cdot e^{-b \cdot \mu} + c) d\upsilon \tag{3}$$

where $a$, $b$, and $c$ are the fitted coefficients, given by the LBLDIS. For a more concise expression, we here abbreviate the wave number integral of radiance $\int_{690cm^{-1}}^{1390cm^{-1}} I(\upsilon, \mu = 1) d\upsilon$ as $II_{\text{AW}}$. Therefore, the final flux calculation function could be written



as follows:

$$\text{LWD}_{\text{AW}} = C \cdot \pi \cdot II_{\text{AW}},$$
$$C = 2 \cdot \int_0^1 (a \cdot e^{-b \cdot \mu} + c)\mu d\mu. \tag{4}$$

$C$ is the correction coefficient for non-isotropic emissions, which is variable for different emissions, such as aerosols, atmo-
sphere in clear day and thin clouds. This correction coefficient $C$ has been determined by LBLDIS model simulations and a
value of 1.35 $\pm 0.05$ for aerosols in the atmospheric window. The method of how to get this correction coefficient is given in
Appendix A.

The error of spectra measured by FTS is usually less than $1\,\text{mWm}^{-2}\text{cm}^{-1}\text{sr}^{-1}$ in AW region (Ji et al., 2023), the uncertainty
of the correction coefficient for non-isotropic emission from aerosol is about $\pm$ 0.05 here, therefore, the theoretical error of
LWD$_{\text{AW}}$ from FTS is $\sqrt{\left(\dfrac{\partial \text{LWD}_{\text{AW}}}{\partial II_{\text{AW}}} \cdot \Delta II_{\text{AW}}\right)^2 + \left((\dfrac{\partial \text{LWD}_{\text{AW}}}{\partial C} \cdot \Delta C)\right)^2}$, about $0.550\,\text{Wm}^{-2}$.

## 4.2 ARE$_{\text{AW}}$ from LBLDIS Model Simulation

To analyze the key parameters affecting the aerosol infrared radiation, we perform model simulations also in the atmospheric
window. Considering model simulation of downwelling emission from the atmosphere, two radiative transfer models are cou-
pled and used in this case, one is the Line-by-Line Radiative Transfer Model (LBLRTM) (Clough et al., 2005) for the gaseous
contribution, another is the DIScrete Ordinate Radiative Transfer model (DISORT) (Stamnes et al., 1988) for calculation of
water droplets and aerosol particles. The coupled model is called LBLDIS (Turner, 2005). This radiative transfer model is
also used as a forward model in the aerosol composition retrieval algorithm described in Ji et al. (2023). The software of the
retrieval algorithm is also public available (see section "dataavailability").

The ARE$_{\text{AW}}$ calculation method from simulated spectra using the LBLDIS model is similar to the method mentioned in
Sec.4.1. The only difference in this section is that additional aerosol information is added to LBLDIS to get the model simulated
ARE$_{\text{AW}}$. ARE of two aerosols, sea salt and sulfate (Ammonium sulfate), are simulated by radiative transfer model (LBLDIS).
Since the model setups for sea salt and sulfate are similar, only the parameters in the model for sea salt are described in
detail here. Usually, aerosol sizes in the Arctic region are often below 1 µm, according to the measurements of aerosol size
distribution in the Arctic (Asmi et al., 2016; Park et al., 2020; Boyer et al., 2022). Weinbruch et al. (2012) found that sea salt
particles were most abundant in particles larger than 0.5 µm. Therefore, in dry conditions, it is assumed that the size of sea salt
is fixed at 1 µm and has the shape of a sphere. The aerosol size distribution is assumed to be a uniform distribution. All aerosols
are fixed at a height of 1000 meters above the ground. Several model simulations are run under various RH conditions (65% as
dry condition, 75% - 95% as wet conditions), with various aerosol number densities ($50\,\text{cm}^{-3}$ - $5000\,\text{cm}^{-3}$). As for sulfate,
the size is assumed to be smaller, 0.4 µm in model simulations in dry conditions, and other settings are the same as those for
the sea salt case. The input data for LBLDIS includes profiles of temperature, pressure, and humidity, which are sourced from
ERA5 (Hersbach et al., 2023).





### 4.3 Aerosol Composition Retrieval from Emission FTS

Ji et al. (2023) describe a modified retrieval algorithm for retrieving aerosol composition. The primary difference in different versions of the retrieval algorithm are the scattering properties look-up tables for various emission sources, such as clouds (Richter et al., 2022) or dry (Ji et al., 2023) or activated aerosols (in this study). For activated aerosols, look-up tables are updated for sea salt and sulfate following the steps described in Ji et al. (2023). An additional step in creating a new look-up table of activated aerosols is to consider the complex refractive index of wet aerosols and the particle size of hygroscopic particles as a function of relative humidity. Therefore, the following parameterization method (Zieger et al., 2013; Petters and Kreidenweis, 2007) is applied:

$$\frac{r_{wet}(RH)}{r_{dry}} = \left(1 + \kappa \frac{RH}{1 - RH}\right)^{1/3} \tag{5}$$

where $r_{wet}$ is the radius of wet aerosols; $r_{dry}$ is the radius of dry aerosols; $\kappa$ is the hygroscopic growth parameter of the aerosols.

To calculate the complex refractive index ($R_{wet} + i \cdot I_{wet}$) of wet aerosol, the volume fraction of dry aerosol (Chin et al., 2002), $f_d$, is used:

$$\begin{cases} R_{wet} = & f_d R_d + (1 - f_d) R_{water} \\ I_{wet} = & f_d I_d + (1 - f_d) I_{water} \end{cases} \tag{6}$$

where $R_d$ and $R_{water}$ mean the real part of the refractive index of dry particles and water respectively; $I_d$ and $I_{water}$ mean the imaginary part respectively.

### 4.4 ARE in the Mid-infrared Range from BSRN

The measurement of LWD (4.5 - 42 μm) from the atmosphere is obtained from the BSRN. Since we are only focusing on the ARE in cloud-free cases in this study, cloud-free conditions should be filtered and radiation from greenhouse gases (GHGs) should be eliminated by the combination of BSRN measurements and radiative transfer simulation based on clear-sky radiosonde data, as follows:

1. Firstly, with the help of the Cloudnet dataset, the LWD measured by BSRN during cloud-free periods is selected, called LWD$_{aero-only}$. Within the altitude range ($0 - 12$ km) of the "Classification" product of Cloudnet (see Sec.3.1), an aerosol-only situation is selected when only "Aerosols" are present in all of the above targets. Then observations of the BSRN are selected that correspond to these times, resulting in aerosol-only BSRN observations. Note that, the LWD from BSRN is the downward radiation of the entire hemispheric atmosphere. This means that simply being cloud-free vertically is not enough to ensure that the entire sky is cloud-free. Therefore, the cloud-free period is ensured to 3 hours around 12:00 (10:30 to 13:30). Besides, during this time period, the radiosonde profile measurement starts at around 11:00 and lasts for about 90 minutes.



2. Water vapor, the most important greenhouse gas (GHG), contributes more significantly to LWD relative to other GHGs (Easterbrook, 2016). Therefore, the next step is to subtract the contribution of water vapor to LWD:

$$ARE = LWD_{\text{aerosol-only}} - LWD_{\text{clear}} \tag{7}$$

where $LWD_{\text{clear}}$ means the infrared radiation flux of clear sky, which is given by the radiative transfer model simulation. Since LBLDIS is mainly used in the AW region, the radiative transfer model used to calculate $LWD_{\text{clear}}$ in mid-infrared range is the SBDART (Santa Barbara DISORT Atmospheric Radiative Transfer, (Ricchiazzi et al., 1998)). Water vapor profiles, pressure, and temperature profiles are from sounding data, other GHGs are fixed to SBDART defaults.

3. Based on the water vapor profiles, we classify the ARE into four scenarios based on the difference in the line shape of the RH profiles: $ARE_{\text{dry}}$, $ARE_{\text{surface}}$, $ARE_{\text{intrusion}}$ and $ARE_{\text{multilayer}}$. $ARE_{\text{dry}}$ means that the entire atmosphere is in a dry state (RH < 60%); $ARE_{\text{surface}}$ means that there is a layer of high humidity (RH > 60%) near the ground (< 1 km); $ARE_{\text{intrusion}}$ represents the situation with a layer of high humidity intrusion (RH > 60%) at high altitude (> 1 km). $ARE_{\text{multilayer}}$ is the case that the atmosphere has multiple layers of high humidity (RH > 60%) and not used in this study.

Note that the time resolution of the $LWD_{\text{aero-only}}$ data is 1 minute, while the time resolution of the sounding profiles is once per day (launched at 11:00 UTC), with each ground to high altitude (about 30 km, maximum height the balloon can reach) observation period lasting about 90 minutes (the sonde needs about 30 minutes to cross the troposphere). Therefore, the $LWD_{\text{aero-only}}$ measurements are averaged over the time period (10:30 UTC - 13:30 UTC) to represent the ARE at 12:00 UTC.

In addition, RH is a key parameter in the aerosol wet growth process and no additional data is indicating at which altitude hygroscopic aerosols are located. On one hand, air masses from the mid-latitudes transport both moisture and aerosols to the Arctic. On the other hand, aerosols are dispersed throughout the atmosphere and become activated at particular altitudes where the relative humidity reaches their deliquescence point. Therefore, we assume here that the peaks in each RH profile are the RH of the activated aerosols. All in all, a total of 100 cases were available after these filter methods were carried out. These cases were divided into four types: 15 cases in $ARE_{\text{dry}}$, 41 cases in $ARE_{\text{surface}}$, 5 cases in $ARE_{\text{intrusion}}$, and 39 cases in $ARE_{\text{multilayer}}$. It is important to note that when there are several peaks of high RH in the atmospheric water vapor profile (here is two peaks in this study), it is challenging to precisely determine the exact layer in which the aerosol resides. To avoid introducing excessive uncertainty, the results for these cases, $ARE_{\text{multilayer}}$, are not included in the results.

Since radiosonde profile measurements are conducted once daily, using the observed atmospheric data to represent conditions between 10:30 and 13:30 may introduce some errors. To assess the uncertainty in the LWD due to daily variations in water vapor, we conducted model simulations. These simulations varied only the water vapor column content while keeping other atmospheric parameters constant, as depicted in Fig. B1. Assuming the profile accurately represents the atmospheric state for half of the time in Ny-Ålesund, while the other half is characterized by subpolar conditions. This assumption results in an LWD effect of approximately $2.8~\mathrm{Wm}^{-2}$. Therefore, it is reasonable to approximate atmospheric conditions during this three-hour window using the once-daily radiosonde profile observations.



## 5    Results

### 5.1    Warming Effect of Aerosols during Wet Growth

When examining the relationship between $ARE_{AW}$ and RH in the model simulation (Sec.4.2), as shown in Fig.1a, a sharp increase in $ARE_{AW}$ is predicted as RH rises. This abrupt enhancement in $ARE_{AW}$ corresponds to the aerosol's deliquescence point. Specifically, the transition point of $ARE_{AW}$ is mainly determined by aerosol composition. For example, the $ARE_{AW}$ associated with sea salt aerosols, characterized by a number density of 500 $cm^{-3}$ (depicted as the solid black line in Fig. 1a), suddenly increases to 10 $Wm^{-2}$ at an RH level of 75%, which is about 5 times higher than that in dry conditions (about 2

$Wm^{-2}$). The magnitude of this number concentration (500 $cm^{-3}$) is within the measurable range at NY-Ålesund (Jung et al., 2018; Pasquier et al., 2022). In contrast, sulfate aerosols exhibit the transition point of $ARE_{AW}$ at 85% as the deliquescence RH of sulfate values at 85% (Peng et al., 2022) (indicated by the dashed black line in Fig. 1a).

Figure 1b depicts the $ARE_{AW}$ measurements under varying ambient RH (Cloudnet is used to determine the altitude at which the aerosol was located, and then the RH value for that altitude is obtained from the ERA5). conditions for different

dominant aerosol composition in Ny-Ålesund based on FTS measurements. When the dominant aerosol is sea salt and sulfate, the $ARE_{AW}$ from the FTS observation also increases as RH rises. The corresponding RH of sudden enhancement of $ARE_{AW}$ in sea salt dominated cases is about 80% - 85%, while that in sulfate dominated cases is around 90%. Among all the observations, three cases with $ARE_{AW}$ exceeded 30 $Wm^{-2}$, and these cases were characterized by sea salt aerosols. Conversely, for non-hygroscopic aerosols, such as dust and black carbon, the $ARE_{AW}$ is about $1.45 \pm 2.00$ $Wm^{-2}$ and does not change with RH.

BSRN measurements give the ARE in the mid-infrared range, as shown in Fig.1c. The analysis in Fig.1c considers the ARE in three distinct scenarios: $ARE_{dry}$, $ARE_{surface}$, and $ARE_{intrusion}$, each representing single-layer high RH scenarios based on water vapor profiles from radiosonde (methods see sec.4.4). Overall, we observe the trending that ARE increases with rising RH. Specifically, under dry conditions (RH < 60%), the ARE remains a low value of about $1.1 \pm 4.4$ $Wm^{-2}$ and does not vary with RH, which is consistent with previous findings (see Fig.1a and b). As RH increases to between 60% and 80%, the

ARE shows a significant increase. Specifically, in the cases of $ARE_{surface}$, the mean ARE averaged between 60% to 80% RH is approximately $19.3 \pm 12.0$ $Wm^{-2}$. As RH exceeds 80%, the ARE escalates rapidly from about 40 $Wm^{-2}$ at 80% RH to approximately 100 $Wm^{-2}$ at 90% RH. Besides, in all five $ARE_{intrusion}$ scenario cases, there are three cases of high water vapor intrusion, but the values of ARE does not increase with RH, and only other two cases show an enhancement of ARE at 70% RH. It's important to note that even under very high ambient humidity conditions (RH > 90%), we still observe low ARE

values, which is due to the presence of non-hygroscopic aerosols (dust or BC) from FTS measurements. Furthermore, within high RH conditions (RH > 90%), there are no intermediate ARE values, with transitions primarily occurring within the RH range of 70% to 80%. This indicates that prevalent hygroscopic aerosols in Ny-Ålesund undergo a transformation from a dry to a wet state within this RH range.





## 5.2 RH Temperature and Sea Salt AOD Changes in the Arctic

To depict the humidity conditions in the Arctic, Figure 2a presents the difference of RH between 2000 - 2022 and 1980 - 2000 in the Arctic at 900 hPa in winter (DJF). In the region around Ny-Ålesund, RH has significantly increased, showing a rise of approximately 2 - 6% compared to pre-2000 levels. Typically, as Arctic warming, rising temperatures often lead to a decrease in RH. However, the notable increase in RH at Ny-Ålesund suggests that specific humidity is increasing more rapidly in this region compared to other parts of the Arctic. This anomaly points to unique local atmospheric conditions or processes that are

enhancing moisture content more effectively than elsewhere in the Arctic.

MERRA2 reanalysis data, illustrated in Figure 2b, indicate a general decrease in sea salt AOD across the Arctic compared to the pre-2000 period. However, there is a notable increase in sea salt AOD in Ny-Ålesund and nearby area. The difference in sea salt AOD between 2000-2022 and 1980-2000 reveals a statistically significant positive anomaly near Ny-Ålesund, approximately +0.005.

When combining the changes in RH and sea salt AOD anomaly, we observe that regions with high humidity and positive sea salt AOD anomalies coincide with areas experiencing large positive temperature anomalies, as shown in Figure 2c. Specifically, these regions exhibit temperature anomalies around +3 $^o$C. These findings highlight the significant role of aerosol wet growth in the Arctic. The suitable RH conditions in Ny-Ålesund have likely facilitated the wet growth of sea salt aerosols, contributing to the observed positive temperature anomalies.

## 6  Discussion

This study has shown that wet aerosols have an additional warming effect. However, at RH levels above 80%, the value of the ARE (approximately 92.7 ±10.7 Wm$^{-2}$) is so high as to be very close to the LWD of clouds (Cox et al., 2015; Serreze and Barry, 2011; Lenaerts et al., 2017; Ebell et al., 2020)). It is well known that aerosols and clouds are closely related. The specific value of the particle size of a wet aerosol that grows to become a cloud droplet particle is not very specific. In this study,

the Cloudnet data is employed for cloud-free filtering. However, our observations reveal that the measured Aerosol Radiative Effect (ARE) values are nearly comparable to the cloud Longwave Downward Radiation (LWD) values at RH levels exceeding 90%. This phenomenon is potentially attributed to the growth of wet aerosol particles into cloud particles after leaving the Ny-Ålesund column. Consequently, as the air mass moves away from the observation site, the newly formed cloud droplets signals become undetectable. This observation aligns with our hypothesis, indicating that the observed ambient RH corresponds to the

deliquescence point, where aerosol particles activate and transition into cloud particles. This finding underscores the reliability of ARE values for RH levels between 60% and 80% compared to those observed at higher RH conditions (> 90%). Given that the maximum ARE value for RH levels between 60% and 80% is approximately 36 Wm$^{-2}$, values exceeding this threshold may be attributed to cloud droplets. In other words, the results of this study indicate that when an instrument is unable to differentiate between the particle sizes of aerosols and cloud droplets with sufficient accuracy, utilizing the ARE or LWD to

distinguish between aerosols and clouds can be a potential method.





Based on the FTS measurements and LBLDIS model simulations, RH and aerosol composition are the most important factors influencing $ARE_{AW}$. The measurement of aerosol composition, especially by remote sensing method, is still challenging. In this study, we applied an FTS retrieval algorithm to retrieve an aerosol composition measurement for all RH conditions, which is a complement to the previous method in Ji et al. (2023). According to Ji et al. (2023), the larger the aerosol AOD, the stronger the aerosol composition signal and the more reliable the retrieved results. For example, in this study, in a case of a sea salt dominant event (compare in Fig.1b), as shown in Tab.1, the AOD of the sea salt is $0.1125 \pm 0.0013$, while that of dust aerosol in this case is $0.0128 \pm 0.007$. BC ($0.0001 \pm 0.0013$) and sulfate aerosol ($0.0001 \pm 0.0109$) are present during this event, however, their contribution is not the dominant factor. In other words, the error of the dominant aerosol composition in AOD retrieval is about 1.16%. Furthermore, as we analysed in sec.4.1, the error of $ARE_{AW}$ measured by FTS is about $0.550 \ \mathrm{Wm^{-2}}$, which is more accurate than that from BSRN (about $5 \ \mathrm{Wm^{-2}}$). Therefore, emission FTS is a helpful instrument to do the aerosol composition and ARE measurements.

Several studies have shown that Arctic water vapor content is increasing, primarily due to enhanced poleward transport from mid-latitudes via atmospheric river pathways (Sato et al., 2022; Thandlam et al., 2022; Bresson et al., 2022; Lauer et al., 2023). ERA5 reanalysis data, as presented in the Fig.2a, indicates that Ny-Ålesund has high water vapor levels during winter, providing suitable ambient RH conditions for aerosol wet growth. MERRA-2 reanalysis data further reveals a positive anomaly in sea salt AOD in the area around Ny-Ålesund after 2000 (Fig.2b). Besides, numerous studies have indicated that during the winter season, sea salt aerosols can dominate the Arctic aerosol composition (Huang and Jaeglé, 2017; Kirpes et al., 2018). Our results (Fig.1b) corroborate this, demonstrating that in scenarios dominated by sea salt aerosols, the infrared radiative effect is most pronounced in Ny-Ålesund. Kirpes et al. (2018) also affirm that sea salt constitutes the principal contributor to accumulation and coarse-mode aerosols during the Arctic winter. Heslin-Rees et al. (2020) have demonstrated that coarse mode aerosols, primarily originating from sea spray, have exhibited an increase over the last two decades (1999 – 2016) at the Zeppelin Observatory on Svalbard. This observed trend can be attributed predominantly to alterations in air mass circulation patterns, with a higher frequency of air masses originating from the Northern Atlantic region (Pernov et al., 2022).

Our measurements, yielding high ARE of hygroscopic aerosols and very low ARE of non-hygroscopic aerosols, indicates sea salt aerosols are very import for the Arctic warming in winter. MERRA-2 reanalysis data, as shown in Fig. 3, confirms our results that in Ny-Ålesund, sea salt and sulfate aerosols are significantly dominant than other aerosol components in winter. As spring arrives, sea salt AOD begins to decline while dust AOD gradually increases. Arctic dust aerosol primarily originates from natural sources such as desert regions, with approximately 65% from Africa (Sahara desert), 22% from Asian deserts, and 13% from other deserts (Kok et al., 2021; Breider et al., 2014). Arctic dust aerosol concentrations peak in spring when long-range transport from Africa and Asia is most efficient (Groot Zwaaftink et al., 2016). This trend continues into summer, when sea salt levels reach their lowest point. Given that sea salt is a major component of winter aerosols, its contribution to Arctic winter warming requires further investigation in the future.

Combining the changes in RH and the sea salt AOD anomaly, the region of high humidity with positive sea salt AOD anomalies overlaps the regions with the large positive temperature anomalies (see Fig.2c). Based on these findings, it is crucial to consider the warming effect of aerosols under high humidity conditions when studying Arctic amplification. The greenhouse





effect of water vapor intensifies surface warming, leading to higher humidity levels in the Arctic (Beer and Eisenman, 2022), which facilitates aerosol wet growth. Increased aerosols in a wet state further warm the Arctic atmosphere, potentially leading to more water vapor and creating a positive feedback loop in Arctic amplification. Therefore, studying the longwave radiation (LWD) contributions from both water vapor and aerosols together is essential.

Although the area of positive sea salt AOD anomalies is very limited in the Arctic, the differences in the deliquescence and efflorescence points during aerosol wet growth process can have the potential to expand the warming effect of aerosols throughout the whole polar regions. Wet aerosols can maintain their hydrated state until they either develop into cloud droplets in supersaturated conditions or revert to dry particles in drier environments, typically below the efflorescence point (Lillard et al., 2009). For instance, as shown in Fig.4, sodium chloride (NaCl) has a deliquescence point of approximately 75% RH

and an efflorescence point of around 46% RH (Peng et al., 2022). Consequently, NaCl remains in a wet state after activation (the black dashed line in Fig.4) until ambient RH drops below the efflorescence point (e.g. 45% RH). Therefore, the Ny-Ålesund region can act as a "refueling station" for the wet growth of the hygroscopic aerosols, specifically the sea salt aerosols, resulting in a warming effect. Here, these aerosols are activated, and after leaving the region, they can remain activated and travel throughout the Arctic, carrying high values of ARE as long as the ambient humidity is higher than the efflorescence

point.

## 7   Conclusions

In this study, based on the measurements from FTS, BSRN, Cloudnet, and radiosonde, the infrared radiative effect of aerosols during the wet growth process has been investigated. Under dry conditions (RH < 60%), the ARE in the whole mid-infrared range remains to about $1.06 \pm 4.35$ Wm$^{-2}$. As RH increases, a significant increase in ARE is observed. Between RH levels

of 60% and 80%, the average ARE is about $19.3 \pm 12.0$ Wm$^{-2}$, about 10 times than that of dry aerosols. Moreover, in cases where the aerosol layer becomes more humid (RH > 80%), the ARE can further increase to about $92.7 \pm 10.7$ Wm$^{-2}$. Besides, the prevalent hygroscopic aerosol in Ny-Ålesund from FTS measurements is sea salt, undergoing a transformation from a dry to a wet state within the RH range of 70% - 80%. The analysis of ERA5 data indicates that Ny-Ålesund has maintained RH levels above 80%, which are conducive to the wet growth of aerosols. MERRA2 reanalysis data shows a positive anomaly in

sea salt AOD in this area, approximately + 0.005, compared to the pre-2000 period. Combining the RH and sea salt AOD, the study finds that areas with high humidity and increased sea salt AOD overlap with regions experiencing significant positive temperature anomalies. Furthermore, if aerosols are highly activated in Ny-Ålesund, they will remain activated after leaving the region as long as the ambient humidity is above the efflorescence point and will propagate throughout the Arctic with high values of ARE.

The results highlight the importance of aerosol wet growth in influencing Arctic climate. The high humidity in Ny-Ålesund as well as nearby area around Svalbard likely promotes the growth of sea salt aerosols, which in turn may contribute to warming through increased longwave downward radiation. These interactions are crucial for understanding Arctic amplification. Continuous monitoring and detailed analysis of these factors are essential for predicting future changes in the Arctic environment.



*Data availability.* All data used in this article are given in detail in sec. 2. Here we briefly illustrate the data. The Ny-Ålesund radiation
measurements are available at the PANGAEA data repository at https://doi.org/10.1594/PANGAEA.914927 (Maturilli, 2020). Data from
Cloudnet (https://cloudnet.fmi.fi/, Ebell et al. (2023)), a product named "Classification", is used to do aerosol-only case selection in BSRN
data. The homogenized radiosonde record obtained is made available at https://doi.org/10.1594/PANGAEA.961203 (Maturilli and Dün-
schede, 2023). The latest version of TCWret (the retrieval algorithm for FTS), including LBLDIS download instructions, can be downloaded
from Zenodo (https://doi.org/10.5281/zenodo.3948048, Richter et al. (2022)). The ERA5 data used in Fig.2 a and c is from Hersbach et al.
(2023). Sea salt AOD data from MERRA-2 can be download at https://disc.gsfc.nasa.gov/datasets/M2TMNXAER_5.12.4/summary (Gelaro
et al., 2017).

## Appendix A:  Non-isotropic emissions correction coefficient

The relationship between integral calculation of radiance in the AW region, $II_{AW}$, with the cosine of the zenith angle, $\mu$ , is
assumed as exponent. Figure A1 presents the relationship $II_{AW}$ with $\mu$. The integral of the fitted function with $\mu$ could be
calculated as the correction coefficient in Equation 4 after getting the fitted function (black dotted lines), and the flux in the
unit of $\mathrm{Wm}^{-2}$ could then be obtained. In this figure, the aerosol type is dry sea salt with a 1 μm (diameter). The method of
aerosol hygroscopic growth mentioned in Sec.4.3 is used to calculate the wet particles. The varied colors indicate the various
number densities of sea salt, while the black line stands for a clear sky scenario. Four cases from the LBLDIS simulation at
four RH conditions (65%, 75%, 85%, and 95%) are provided as well. When the aerosol number density is low, between 50
and 500 $\mathrm{cm}^{-3}$, the ratio of $II_{AW}(\mu)$ to $II_{AW}(\mu = 1)$ at various u remains relatively constant. With a sharper relationship
in dry conditions and a progressive flattening with an increase in RH, this phenomenon occurs in all RH cases. Furthermore,
the differences in the equations at higher relative humidity levels (75%, 85%, and 95%) are not apparent, suggesting that the
correction coefficient for non-isotropic aerosol scenarios may be similar under wet conditions. Additionally, such a relationship
dramatically flattens as RH rises in the presence of heavy aerosol pollution, such as that present in the case of 5000 $\mathrm{cm}^{-3}$, and
tends to be isotropic in high RH as in the case of RH=95%. We anticipate that under high RH condition, a significant number
of aerosols will progressively activate and develop into a thick cloud in the atmosphere that emits isotropic radiation.

Following the simulation of the sea salt, the sulfate aerosol is also calculated by LBLDIS. Table .A1 displays the whole
results, including the atmosphere on a clear day. The correction coefficient for a clear atmosphere is approximately 1.08. For
a clear sky, the light from atmosphere emission could be very nearly isotropic. While for aerosols, sea salt has a correction
coefficient of 1.40 in dry state and roughly 1.35 in wet, which is considerably different from clear day. Although the activated
RH for sulfate aerosol and sea salt is different (75% for sea salt and 85% for sulfate), both exhibit similar correction coefficient
values when they are activated.

In conclusion, non-isotropic radiance emission from aerosols should be taken into account while doing flux calculation for a
moderate aerosol event, which frequently occurs in the Arctic. In dry conditions, aerosols have a different correction coefficient
than they do in wet conditions. It is not very varied for various aerosol types.



*Author contributions.* D.J., X.S., M.P.,M.B. and J.N. conceived and designed the study. D.J. collected, organized and processed data, developed the retrieval algorithm and write this paper. K.E. gave advice in LWD from water vapour and provided detailed descriptions of the Cloudnet data. M.M. gave advice in BSRN and sounding data. X.S. gave advice in model simulation, data processing and physical mech-anism. M.P. and M.B designed and built the measurement setup, performed FTS measurements and gave advice in the retrieval algorithm.
J.N. gave advice in the article structure and the frame of the scientific content. All co-authors made contribution to the revision of this article.

*Competing interests.* The authors declare no competing interests.

*Acknowledgements.* We acknowledge ACTRIS and Finnish Meteorological Institute for providing the data set which is available for download from https://cloudnet.fmi.fi. The cloud radar data for Ny-Ålesund was provided by the University of Cologne, the ceilometer and microwave radiometer data by the Alfred Wegener Institute, Helmholtz Centre for Polar and Marine Research. We thank the staff of AW-
IPEV research base in Ny-Ålesund for technical support of the measurements. This work is funded by the AWIPEV research as part of the projects AWIPEV-0016, and the Deutsche Forschungsgemeinschaft (DFG, German Research Foundation) - project number 268020496 - TRR 172, within the "Transregional Collaborative Research Center 'ArctiC Amplification: Climate Relevant Atmospheric and SurfaCe Processes, and Feedback Mechanisms (AC)3'". We acknowledge ECMWF for providing IFS model data, DWD for providing ICON model data, and NCEP (National Centers for Environmental Prediction) for providing access to GDAS1 data. We thank the AWI Bremerhaven and
the AWI Potsdam for logistical support on the AWIPEV research base and the station personnel for on-site support. We thank the senate of Bremen for partial funding of this work.



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





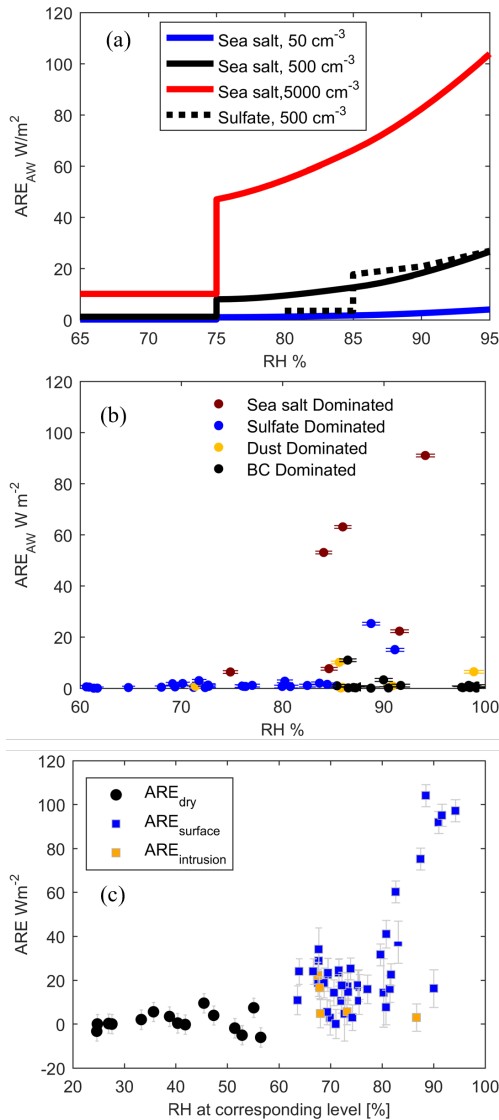

**Figure 1.** (a) Aerosol Radiation Effect (ARE$_{AW}$) of sea salt (red, black and blue lines) and sulfate (black dotted line) as a function of RH, simulated by LBLDIS with different number density cases; (b) The ARE$_{AW}$ of sea salt(brown), sulfate (blue), dust(yellow) and BC (black) dominant cases measured by emission FTS (NYAEM-FTS). The aerosol composition retrieval method is given in Sec.4.3 and the methods is given by Ji et al. (2023); (c) ARE under different RH profile scenarios: ARE$_{Dry}$ (black) means that the entire atmosphere is in a dry state (RH < 60%); ARE$_{surface}$ (blue) means that there is a layer of high humidity (RH > 60%) near the ground (< 1 km); ARE$_{intrusion}$ (yellow) represents the situation with a layer of high humidity intrusion (RH > 60%) at high altitude (> 1 km). The error bars represent one standard deviation of the ARE calculated over a 3-hour period (10:30 - 13:30). Note: ARE$_{AW}$ in this figure (a) refers to simulations and (b) refers to measurements by NYAEM-FTS in the AW region, and ARE in figure c refers to the results of measurements (BSRN) in the mid-infrared range.



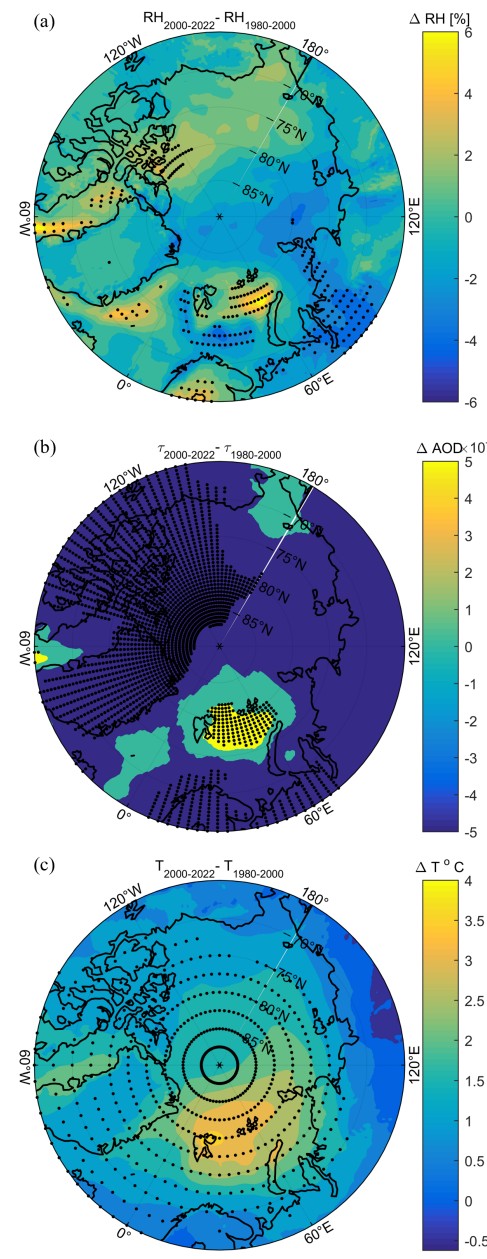

**Figure 2.** (a) The difference of RH between 2000 - 2022 and 1980 - 2000 in the Arctic at 900 hPa in winter (DJF), data from ERA5 (Hersbach et al., 2023); (b) The difference of Sea salt aerosol optical depth between 2000 - 2022 and 1980 - 2000, data from Merra-2 reanalysis data (Gelaro et al., 2017); (c) The difference of temperature between 2000 - 2022 and 1980 - 2000 in the Arctic at 900 hPa in winter (DJF), data from ERA5 (Hersbach et al., 2023). The black dots in (a), (b), and (c) mean the difference of this grid passes the significance test (95%).



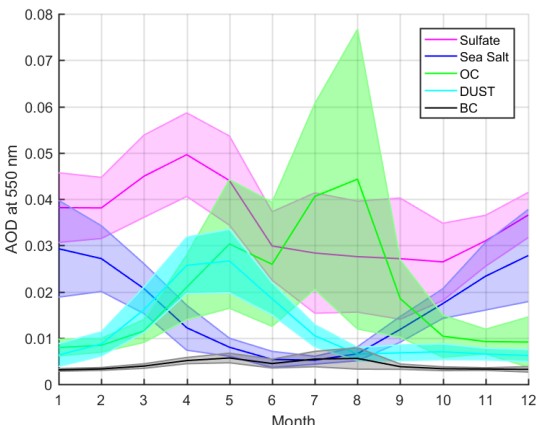

**Figure 3.** Seasonal variation of sulfate, sea salt, OC, dust and BC from MERRA-2 reanalysis data averaged from 2002 to 2021 with one standard deviation (shaded area) in Ny-Ålesund.





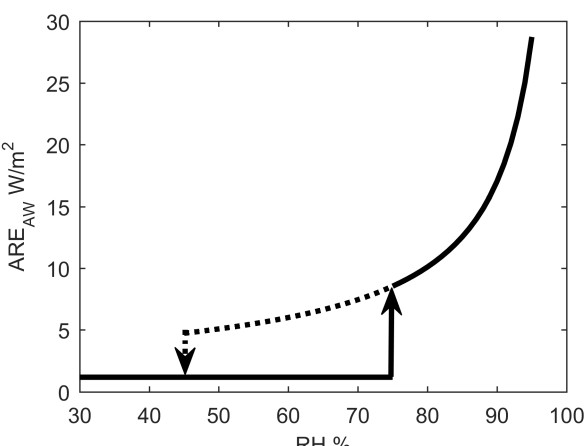

**Figure 4.** Graph depicting Aerosol Radiative Effects (ARE) as a function of relative humidity (RH), accounting for the deliquescence and efflorescence points. The aerosol component illustrated is sodium chloride, with an aerosol number concentration of 500 particles per cubic centimeter. The deliquescence point is set at 75%, and the efflorescence point is at 46%.



**Table 1.** Retrieval results (AOD and error) from an aerosol event dominated by sea salt.

| Aerosol composition | sea salt | Sulfate | BC | Dust |
|---|---|---|---|---|
| AOD | 0.1125 | 0.0001 | 0.0001 | 0.0128 |
| Error in AOD | 0.0013 | 0.0109 | 0.0013 | 0.0070 |



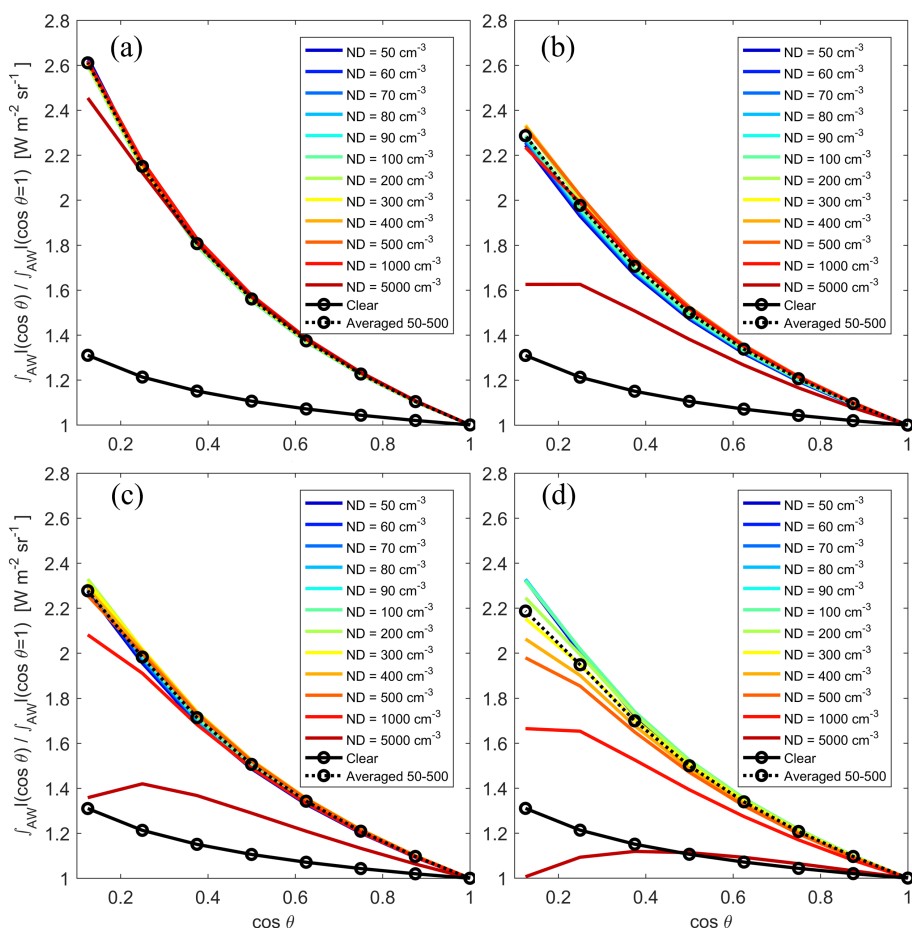

**Figure A1.** (a)The relationship between $\mathrm{II}_{\mathrm{AW}}$ with $\mu$. The varied colors indicate the various number densities of sea salt, while the black line stands for a clear sky scenario. The black dotted lines is the averaged from 50 $\mathrm{cm}^{-3}$ to 500 $\mathrm{cm}^{-3}$. Four cases from the LBLDIS simulation at four RH conditions (65% in (a), 75% in (b), 85% in (c), and 95% in (d)) are provided as well.





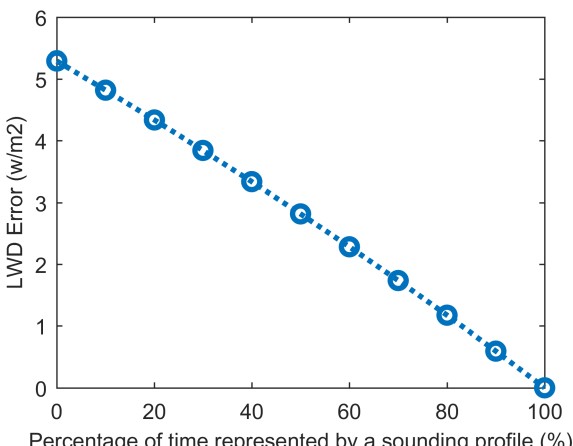

**Figure B1.** Errors in LWD due to uncertainty in the proportion of time that can be represented by the 90-minute profile over a three-hour period, simulated using the SBDART. The model simulation for Longwave Downward Radiation (LWD) involves changing only the water vapor column content, keeping other atmospheric conditions constant. For example, assuming the profile observations (90 minutes) are representative of only half of the 3-hour period, we consider the scenario: the profile accurately represents the atmospheric state (0.3 $gcm^{-2}$(Pałm et al., 2010)) for half of the time in Ny-Ålesund, while the other half is characterized by subpolar conditions (0.42 $gcm^{-2}$, model default). This scenario results in an LWD effect of approximately 2.8 $Wm^{-2}$.





**Table A1.** The correction coefficient for atmosphere, sea salt and sulfate at different RH.

| RH | <75% | 75% | 85% | 95% |
|---|---|---|---|---|
| C_Atmos | 1.08 | 1.08 | 1.08 | 1.08 |
| C_sea salt | 1.40 | 1.35 | 1.35 | 1.35 |
| C_sulfate | 1.36 | 1.36 | 1.37 | 1.33 |