# Peer review of "Hygroscopic Aerosols Amplify Longwave Downward Radiation in the Arctic"

_EGUsphere, 2024_

## Author Comment (AC1)

Response to Comments of reviewer 1

The authors thank both reviewers for their constructive comments and suggestions, which have helped us to improve the quality of this paper both in sciences and writing. The response in black letters follows each comment in blue.

**General**

This manuscript investigates the roles of hygroscopic aerosols in Arctic longwave downward radiation. This is an interesting and potentially important topic as the Arctic is changing rapidly. Most of the methods and analyses are reasonable and clearly presented. My only concern is the impact of potential cloud scenes on the estimated ARE – the numbers are high for RH > 80% and almost comparable with the cloud effect. This finding is significant if the signals are truly from wet aerosols. However, further examination of the measurement data is needed to exclude cloud contamination, especially from nearby regions around the surface site. The manuscript is otherwise well written, and I recommend returning it to the authors for minor revision. A major revision is appropriate if substantial work is needed to examine the measurements.

Response:

Based on the reviewer's concern about the "potential cloud scenes", we first checked all the data and method we used, including FTS observation and BSRN dataset. After updating the dataset, the aerosol infrared radiative effect (ARE) induced by aerosol wet growth remains evident, and the magnitude is more reasonable compared to the earlier version. The detailed updates are as follows:

**1. FTS Observations:**

Initially, given the small field of view (FOV) of the FTIR instrument (3.3 mrad), we focused on vertical cloud screening, using our Cloudnet-radar, assuming this would be sufficient. In the Arctic, where cloud heights are typically assumed to be around 1 - 5 km, the FOV at this altitude corresponds to a radius of approximately 8.25 m. While the movement speed of high-altitude clouds is uncertain, we removed any data associated with potential cloud signals detected within 30 minutes before and after the observation time. This approach ensures that the FOV is cloud-free. With this stricter screening, the consistency between the FTIR observations and model simulations improved. The model-simulated sea salt concentration was approximately 500 particles/cm³, a value consistent with known Arctic observations.

Based on these updates, we have added the following sentences:

(L153-155) "Considering the small field of view (FOV) of the FTIR instrument (3.3 mrad), we excluded any data with cloud signals were detected within 30 minutes before or after the observation time. This method ensures that the FTIR's FOV remains cloud-free during the analysis."

**2. BSRN Observations:**

For the BSRN data with all-sky observation (FOV = π rad), we considered the possibility that cloud contamination might persist even if no clouds were detected by Cloudnet within 90 minutes before and after 12:00 UTC. To address this, we introduced an additional criterion: if cloud signals from Cloudnet were detected outside this 180-minute window, the corresponding data were flagged as cloud-contaminated and excluded from our aerosol radiative effect (ARE) calculations. This stricter criterion ensures that the remaining data represents whole day cloud-free sky conditions.

However, we acknowledge that this approach might be overly conservative for aerosols. For example, thick clouds located near the horizon, despite their optical thickness, contribute less to the overall signal due to the high zenith angle ($\sin(\theta) \rightarrow 0$). In such cases, aerosols at the zenith might still dominate the observed ARE.

We now provide a figure as a schematic illustration of our cloud contamination search. We also added the following sentences:

(L254-258)"Besides, we also consider the possibility that cloud contamination might persist despite the absence of detectable clouds within the 180-minute window before and after 12:00 UTC. To address this consideration, an additional criterion is implemented: data are flagged as cloud-contaminated and excluded from the aerosol radiative effect calculations if cloud signals were detected by Cloudnet outside this 180-minute window. This enhanced screening method ensures that the remaining dataset represents cloud-free sky conditions."

[Figure]

Figure 1: A schematic illustration of the cloud contamination.

**Minor**

L151-153: what is the field of view of FTS? Do you restrict the area coverage that needs to be cloud-free from Cloudnet? Multiple scattering could contribute to the FTS measurement if cloud is present in nearby regions.

Answer: The field of view (FOV) of the FTIR is 3.3 mrad. In the new version of our manuscript, we have added the following sentences (mentioned before when response to the major comments):

 (L153-155) "Considering the small field of view ... remains cloud-free during the analysis".

L160: please define LBLDIS

Answer: The LBLDIS model is the Line-By-Line Radiative Transfer Model coupled with DIScrete Ordinates Radiative Transfer Model. In the new version of our manuscript, we have added this information:

(L164-166)" ..., which can be calculated using the LBLDIS model (Line-by-Line Radiative Transfer Model coupled with DIScrete Ordinates Radiative Transfer Model, and details of this model are given in the Sec.4.2.) "

Figure 1: The ARE from measurements for sea salt-dominated scenarios (b) can be comparable to 5000/cm3 of sea salt in simulations. Is this a reasonable sea salt number concentration in the Arctic winter? This raises a concern about whether the sky is truly cloud-free in observations.

Answer: 5000 particles/cm³ is indeed too high to be practical in the Arctic. This is only a theoretical calculation and is not comparable to observations. In our modified version, we removed the case of 5000 particles/cm³ from the new figure 1. With this stricter screening method applied to FTIR as mentioned in the response to the major comments, the consistency between the FTIR observations and model simulations improved. The model-simulated sea salt concentration was approximately 500 particles/cm³, a value consistent with known Arctic observations. We have updated the data in the new version, see Figure 1, and for convenience we also present this new version at the end of the reply. In the new version of our manuscript, we have added this information:

(L286-291)"Moreover, the FTIR observations align closely with model simulations ( sea salt case with a number concentration of 500 cm$^{-3}$). Both observations and simulations show that during the early stages of aerosol wet growth (75% < RH < 80%), the aerosol infrared radiative effect increases from approximately 1 – 2 Wm$^{-2}$ to 10 Wm$^{-2}$. Subsequently, as RH approaches 90%, the ARE reaches about 20Wm$^{-2}$."

Figure 2: there is a white line at 180. This might be because the lat/lon do not overlap or connect in the array for 180E and 180W. To get rid of this, you can manually add an extra column in your data array. Suppose you have a [90, 360] shaped data, you add the 361st column that is identical to the 1st column and, correspondingly, your coordinate arrays. Then, you plot the [90, 361] shaped array. This should make the contour connected.

Answer: Corrected.

L321-325: then do you still consider the ARE values at RH levels above 80% valid for aerosols (e.g., the beginning of Section 6)? And I assume the RH here is with respect to liquid, right? During the Arctic winter, RH of 80% with respect to liquid could potentially be high enough for ice cloud formation. Distinguishing cloud and aerosol conditions is vital for the estimated ARE for hygroscopic aerosols.

Answer: The RH referenced here is relative to liquid water. We agree that an RH of 80% relative to liquid water under Arctic winter conditions could potentially be high enough for ice cloud formation. However, it is important to note that the formation of ice clouds generally requires specific ice-nucleating particles (INPs), such as dust or black carbon aerosols, rather than hygroscopic aerosols like sea salt or sulfate. For the transformation from supercooled liquid droplets to ice clouds, extremely low temperatures (e.g., below -38°C) are typically required. In the temperature from 0°C to -38°C, hygroscopic aerosols are likely to remain in the liquid phase and act as supercooled droplets rather than forming ice clouds. Therefore, it is reasonable to treat hygroscopic aerosols as liquid droplets under these conditions, and their additional radiative effect (ARE) should still be considered valid.

In the new version of our manuscript, we have added the following information: (L310-314) Notably, we differentiate the potential radiative effect from the cloud contamination. In Fig.1c, when RH is 60% - 80%, the infrared radiative effect from aerosol only (orange and blue dots) can be about 20 Wm$^{-2}$, which is comparable with the infrared radiative effect combined aerosol with potential cloud contamination (gray dots). However, when the environment becomes more humid (RH > 80%), differentiating the radiative effect between cloud and aerosol is challenging due to the observation method. This implies that the estimation of ARE with RH less than 80% is more reliable than that of RH>80%.

We added the following sentences in the discussion:

(L343-352)"Our study shows that wet aerosols have an additional warming effect. However, when the relative humidity exceeds 80%, Cloudnet always observes some cloud signals during the period beyond the 180-minute window in BSRN measurements. We cannot conclusively determine whether high values of LWD (> 40Wm$^{-2}$) are solely caused by aerosols or are the result of cloud contamination (see Fig.1). Under the very humid conditions (RH > 80%), wet aerosols become activated and transform to cloud droplets. This phenomenon from BSRN observation aligns with our hypothesis, indicating that the observed ambient RH corresponds to the deliquescence point, such as 80% for sea salt (Peng et al., 2022). Given that the maximum ARE value for RH levels between 60% and 80% is approximately 36 Wm$^{-2}$, values exceeding this threshold may attribute to cloud droplets. The results of this study indicate that when an instrument is unable to differentiate between the particle sizes of aerosols and cloud droplets with sufficient accuracy, utilizing the ARE or LWD to distinguish between aerosols and clouds can be a potential method. "

Figure 2. (New version of Figure 1 in re-submission manuscript) has been modified as shown below. The model simulation for the 5000 cm⁻³ case has been excluded. Figure (b) and (c) have been updated because of cloud signals.

[Figure]

(a) Aerosol Radiation Effect (ARE$_{AW}$) of sea salt (red, black and blue lines) and sulfate (black dotted line) as a function of RH, simulated by LBLDIS with different number density cases; (b) The ARE$_{AW}$ of sea salt(brown), sulfate (blue), dust(yellow) and BC (black) dominant cases measured by emission FTIR (NYAEM-FTS). The aerosol composition retrieval method is given in Sec.4.3 and the methods is given by Ji et al. (2023); (c) ARE under different RH profile scenarios: ARE$_{Dry}$ (black) means that the entire atmosphere is in a dry state (RH < 60%); ARE$_{surface}$ (blue) means that there is a layer of high humidity (RH > 60%) near the ground (< 1 km); ARE$_{intrusion}$ (yellow) represents the situation with a layer of high humidity intrusion (RH > 60%) at high altitude (> 1 km). Grey crosses indicate cloud contamination. The error bars represent one standard deviation of the ARE calculated over a 3-hour period (10:30 - 13:30). Note: ARE$_{AW}$ in this figure (a) refers to simulations and (b) refers to measurements by NYAEM-FTS in the AW region, and ARE in figure c refers to the results of measurements (BSRN) in the mid-infrared range.

---

## Author Comment (AC2)

Response to Comments of reviewer 1

The authors thank both reviewers for their constructive comments and suggestions, which have helped us to improve the quality of this paper both in sciences and writing. All comments are carefully considered and responded to. The response in black letters follows each comment in blue.

The authors' investigation into the amplification of longwave downward radiation by hygroscopic aerosols in an Arctic field site presents notably high radiative effect values. This Referee maintains a degree of skepticism and recommends a revision of the manuscript to address these concerns convincingly. The following points should be considered:

- **Comparison with Existing Studies:** Given the dramatic results, it's essential to compare and contrast these findings with existing literature. Are there any reported measurements or theoretical calculations in similar or relevant settings?

Answer: To our knowledge, no observations have been reported of additional infrared radiation released during aerosol wet growth. We hope that our observations will contribute to this gap. Under the condition of dry aerosol particles, our results are consistent with several literatures. And we have added the following sentences:

(L292) "Conversely, for non-hygroscopic aerosols, such as dust and black carbon, the $ARE_{AW}$ is about $1.45 \pm 2.00$ $Wm^{-2}$, and close to previous studies, which does not change with RH (Spänkuch et al., 2000; Markowicz et al., 2003; Vogelmann et al.,2003; Lohmann et al., 2010)."

- **Robustness of Results:** In line with Referee 1's comments, additional effort is needed to ensure the robustness of the results. For example, while the authors discuss the distinctions between dry aerosol particles, wet aerosol particles, and cloud condensation nuclei, further elaboration is necessary.

Answer: Our study focuses on humidity levels below 100%, meaning we only discuss aerosols in their dry and wet states. We take aerosol in RH < 60% as dry states. When the environment becomes more humid (RH > 60%), a hygroscopic particle can absorb water, and its size grows, which can act as cloud condensation nuclei (CCN). This hygroscopic particle is defined as wet aerosol in our study. It is worth noting that since we focused on the infrared radiation effect of aerosols, the most notable distinction between the dry and wet states is that aerosols in the wet state (RH > 60%) exhibit enhanced infrared radiation.

And we added the following sentences to the introduction: (L54 - 57)" Our study focuses on humidity levels below 100%, meaning we only discuss aerosols in their dry and wet states. We take aerosol in RH < 60% as dry states. When the environment becomes more humid (RH > 60%), a hygroscopic particle can absorb water, and its size grows, which can act as cloud condensation nuclei (CCN). This hygroscopic particle is defined as wet aerosol in our study."

**Clarity of Methods:** The Methods section requires additional clarity. For instance, the introduction of "AREaw from FTS" measurements in section 4.1 that uses LBLDIS model calculations before the latter's formal introduction in the subsequent 4.2 section, creates ambiguity. The source of the evidence should be explicitly stated.

Answer: Corrected, we define the model in Section 4.1.(L164-166).

- **Precision in Terminology and Notation:** Equation 1 and its description warrant careful attention. The equation defines AREaw as the difference between all-sky and clear-sky values, yet the description refers to all scenes as "clear-sky" (no clouds), with the "clear-sky" term in the equation implying the absence of both clouds and aerosols. A revised notation and convention, possibly using the term "clean" for scenarios without aerosols, could enhance clarity.

Answer: Corrected, we have changed all the "clear-sky" to "clean-sky" referring to the conditions without aerosols and clouds.

- **Methodological Clarity and Validation:** The relationship between the various radiation methods introduced needs clarification. Are they complementary or intended for cross-checking? Additionally, the manuscript would benefit from a discussion of any validation efforts undertaken to bolster confidence in the results.

Answer: Added in the discussions part:

(L331 - 342) "FTIR and BSRN observations are operating on different spectral bands—FTIR focusing on the atmospheric window spectrum region and BSRN covering a broader infrared spectrum. It is worth noting that the estimation of the absolute radiation value from two observation methods is not comparable because of the different spectral range. However, if the cross-validation of these methods is needed, we can roughly compare them in terms of how many times they have grown in radiation from dry to wet aerosol. Both FTIR and BSRN observations consistently indicate that within the relative humidity range of 60% – 80%, aerosol wet growth results in an approximate 7 times increase in ARE compared to dry conditions. At high humidity (> 80%), the FTIR instrument can capture the infrared radiative enhancement by aerosol wet growth because of the small field of view (FOV = 3.3 mrad). In contrast, BSRN all-sky observation, which requires a completely cloud-free sky across the entire observation domain, is more susceptible to cloud contamination under high-humidity conditions. As a result, BSRN is limited in providing precise ARE values at higher humidity levels. This distinction highlights the strengths and limitations of each observational method under different atmospheric conditions."

More general comments that could be considered in the revision as well:

- **Extrapolation and Temperature Impact:** The manuscript's impact could be enhanced by extrapolating the local effects to a larger (regional or global) signal regarding longwave radiation effects. Furthermore, can the measurements provide

insights into the contribution of these aerosol effects to observed temperature changes nearby?

Answer: We acknowledge the importance of extrapolating the local effects observed in this study to larger regional or global scales to better understand the broader implications of aerosol longwave radiation effects. We are currently conducting model simulations for the recent 10 years to explore these regional and global impacts. However, as model simulations are very time consuming, we are still in the process of obtaining the first results. We plan to update this part of the work in future studies once the simulations are complete. These model results will also allow us to investigate the potential contribution of aerosol effects to observed temperature changes in the surrounding regions. We greatly appreciate your suggestion and have noted this as a key direction for future research.

- **Writing Style and Clarity:** The manuscript's readability could be improved. The use of numerous acronyms, while potentially common in this subfield, can hinder comprehension. Careful consideration of whether each acronym is necessary would enhance clarity. For example, the authors use "FTS" but maybe "FTIR" is more appropriate here?

Answer: Corrected. In the new version of our manuscript, we have used commonly recognized abbreviations wherever possible and minimized the use of unnecessary abbreviations.